

# Assessing the growth in clinical skills using a progress clinical skills examination

Heather S. Laird-Fick[1], Chi Chang[2], Ling Wang[1], Carol Parker[3], Robert Malinowski[2], Matthew Emery[4] and David J. Solomon[5]

[1] Department of Internal Medicine, Michigan State University, East Lansing, MI, USA
[2] Office of Medical Education Research and Development/Department of Epidemiology, Michigan State University, East Lansing, MI, USA
[3] Academic Affairs and Office of Medical Education Research and Development, Michigan State University, East Lansing, MI, USA
[4] Department of Emergency Medicine, Michigan State University, Grand Rapids, MI, USA
[5] Department of Internal Medicine/Office of Medical Education Research and Development, Michigan State University, East Lansing, MI, USA

Corresponding author
David J. Solomon,
dsolomon@msu.edu

## ABSTRACT

**Background:** This study evaluates the generalizability of an eight-station progress clinical skills examination and assesses the growth in performance for six clinical skills domains among first- and second-year medical students over four time points during the academic year.

**Methods:** We conducted a generalizability study for longitudinal and cross-sectional comparisons and assessed growth in six clinical skill domains via repeated measures ANOVA over the first and second year of medical school.

**Results:** The generalizability of the examination domain scores was low but consistent with previous studies of data gathering and communication skills. Variations in case difficulty across administrations of the examination made it difficult to assess longitudinal growth. It was possible to compare students at different training levels and the interaction of training level and growth. Second-year students outperformed first-year students, but first-year students' clinical skills performance grew faster than second-year students narrowing the gap in clinical skills over the students' first year of medical school.

**Conclusions:** Case specificity limits the ability to assess longitudinal growth in clinical skills through progress testing. Providing students with early clinical skills training and authentic clinical experiences appears to result in the rapid growth of clinical skills during the first year of medical school.

## INTRODUCTION

Progress testing uses broad-based examinations that are designed to assess end of curriculum objectives and are given at regular intervals over a course of study. Progress testing was first implemented in medical education during the 1970s and has grown to be widely used in medical schools and to some extent residency programs (*Albanese & Case, 2016*). There is a considerable body of research that suggests progress testing is

particularly well suited for less structured problem focused curricula and encourages learners to study in ways that promote understanding rather than rote memorization (*Blake et al., 1996*). Progress testing also provides a rich and systematic source of data for student feedback, program evaluation and identifying students that need remediation.

Progress testing has almost exclusively been implemented in the form of written examinations and rarely as clinical skills examinations. We could only identify one systematic implementation of clinical skills progress testing, and it was in an internal medicine residency program (*Pugh et al., 2016*). Many medical schools are now placing first-year students into authentic clinical settings, increasing the potential value of progress clinical skills examinations (PCSEs) for providing systematic feedback on the growth of their students' clinical skills. Furthermore, in the USA, the expectations for medical school graduates entering residency training are being operationalized through Core Entrustable Professional Activities (EPAs) for Entering Residencies (*Association of American Medical Colleges, 2014*) that do not necessarily lend themselves to being assessed via written examinations.

The changes in both the structure of undergraduate medical education and the expectations for graduates has increased the value of PCSE as an integral part of medical school assessment and evaluation programs. Although there has been ample research on the psychometrics, acceptance and impact of written progress testing (*Albanese & Case, 2016*), there has been virtually no research done on PCSEs.

The goal of this study is to assess the psychometric and practical challenges of implementing a progress clinical skills program and our early findings about the growth in clinical skills over the first 2 years of medical school in a curriculum that includes early clinical experiences and extensive clinical skills training. Specifically, this study estimates the generalizability of an eight-station PCSE for assessing longitudinal growth in clinical skills over the course of the curriculum using different standardized patient (SP) cases for each administration of the examination. The study also estimates the generalizability coefficients for assessing cross-sectional differences in student performance at different levels of training using the same SP cases.

Generalizability is a very flexible approach for estimating the reliability or "generalizability" of test scores that is based on analysis of variance (*Tavako & Brennan, 2013*). Rather than just providing a global estimate of the measurement error in scores, it allows the error variance to be partitioned into specific sources or facets. This can be useful allowing educators to not only assess the reproducibility of test scores but also assess what are the most significant sources of the error and in turn the most efficient and effective means of improving the accuracy of the assessment process. That is one of the specific aims of the study.

While there are general guidelines for the reliability or generalizability necessary for various types of testing such as "high stakes" assessment, these guidelines are essentially meaningless when taken out of the context of the measurement situation. As noted in the American Educational Research Association guidelines for high stakes testing (*AERA, 2000*) the reliability of a test or other assessment needs to be sufficient for its intended use of the test scores. This issue is addressed in the discussion section of the

paper both as it relates to our medical school's use of PCSE in student assessment and how it might apply in other medical schools.

A second goal of this study is to assesses the impact of authentic clinical experiences and weekly clinical skills training in a simulation laboratory with faculty feedback on the growth of clinical skills over the first 2 years of medical school. As noted above, more and more medical schools are including authentic clinical experiences in the first 2 years of training. Understanding the impact of these experiences on the growth of clinic skills is increasingly important.

In summary, the study addresses the following research questions.

1. Is the reproducibility of an eight station, approximately 2 hour PCSE examination, adequate for:

   a. comparing the performance of students taking the same examination,
   b. assessing growth over the curriculum for the same students taking multiple examinations using different SP cases.
   c. making decisions about the adequacy of students' clinical competency for progressing in the medical school curriculum.

2. Does authentic clinical experiences and weekly clinical skills training in the first and second year of medical school result in the rapid growth in clinical skills and is that growth sustained through the first 2 years of medical school?

## MATERIALS AND METHODS

### Subjects and setting

In the fall of 2016, Michigan State University's College of Human Medicine implemented a new curriculum called the Shared Discovery Curriculum (SDC). The SDC is organized around patient chief complaints and concerns. Students start their medical training by learning basic data gathering and patient communication skills through simulation-based educational experiences that includes SP encounters and direct observation and feedback from clinicians. After 8 weeks of training, students begin working two half-days a week in clinic settings with medical assistants and nurses. As they gain more experience, their clinical responsibilities grow. At the same time, they work to master clinical applications of basic science knowledge independently, in small groups, and in a weekly large group session. They also continue to receive 4 hours per week of clinical skills training in the medical school's simulation centers.

### Measures

The students are evaluated for both formative and summative purposes via progress testing twice each semester using both written and clinical skills examinations. Descriptions of the development and piloting of the PCSE used in the SDC has been published elsewhere (*Gold et al., 2015*; *DeMuth et al., 2017*). The examination uses an objective structured clinical examination format (*Harden et al., 1975*) and consists of eight 15-minute SP
encounters with 10-minute post encounter stations. Each encounter assesses some combination of patient interaction skills, hypothesis-driven history gathering, physical examination, counseling and safety behaviors using checklist and rating items completed by the SP. While the SPs complete the forms, the examinees complete tasks in a post encounter station which are rated by faculty. Descriptions of what is assessed in each of the domains along with a few illustrative items are provided below.

- **Interactional skills** are assessed with checklist items addressing students' ability to establish and maintain good rapport, put the patient at ease and set an agenda. Checklist items include "greeted me pleasantly," "made regular eye contact," "asked about ALL of my concerns," and "asked how illness is affecting my life."

- **Problem-focused history taking skills** are assessed with checklist items addressing students' ability to acquire information about the patient's medical history and current illness in a logical and orderly fashion. Checklist items include "elicited/asked when symptoms started," "asked me to rate my pain on a scale of 0 to 10," "elicited/asked obstetric or gynecologic history," and "elicited/asked about my current occupation."

- **Problem-focused physical examination skills** are assessed with checklist items addressing students' ability to perform a logical and orderly exam, including the ability to select those parts of the exam most relevant to working through a differential diagnosis appropriate to the chief complaint and history. Checklist items include "checked pupils with pen light or scope," "tapped or pressed directly over my spine," "listened to my abdomen with a stethoscope," and "watched me walk."

- **Patient counseling skills** are assessed with checklist items addressing students' ability to provide counseling appropriate to identified or suspected problems, engage in shared decision-making and obtain informed consent. Checklist items include "provided advice about managing stress," "sat while delivering bad news," "used a decision support aid," and "asked if I was comfortable with the plan."

- **Patient safety skills** are assessed with checklist items addressing students' ability to identify and mitigate common and major safety risks. Checklist items include "wore gloves," "asked for help," "used generic medication names," and "verified medication allergies and reactions."

- **Post-encounter tasks** are graded by faculty members, assess the application of medical knowledge, clinical reasoning, and clinical documentation. Examples include writing a note in chart or developing an appropriate differential diagnosis for the patient.

The PCSE stations are designed to assess EPAs not easily assessed by written examinations. The performance of an examinee is reported as the percentage of possible points students achieved across all eight cases in each of the six domains.

The SPs used in the PCSE are trained to the specific PCSE cases they simulate. Both their portrayal of the case script and the accuracy of their completion of the checklist/rating forms are assessed by the staff in the simulation centers before each PCSE is given and include measurements of inter-rater reliability. Adjustments to either the

case or how the SP is trained are made when these quality assurance efforts identify a problem.

## Design

The PCSE is given as part of a broad-based progress assessment that also includes written examinations. These progress assessments occur twice each semester for a total of 20 assessments over the course of the medical school curriculum. Third- and fourth-year students are assessed using the PCSE once each semester. Depending on their rotational schedule each third- and fourth-year student is assessed either in the first or second PCSE given that semester. To pass in a semester, students must pass at least one of the two PCSE given that semester with scores deemed appropriate for their level of training. Third- and fourth-year students who do not meet course-specific expectations for all skill areas on the PCSE are given an opportunity to take a make-up PCSE as a means of demonstrating their competency.

Since students in all 4 years of training take the same PCSE examination at roughly the same time, we can potentially observe the growth in clinical skills both longitudinally over the course of each students' medical training as well as cross-sectionally between students with different levels of training taking the same PCSE. The SP cases for each PCSE are drawn from a pool of cases that are continually being developed. The SP cases will eventually be reused but only after the students who were originally assessed via the case have graduated. As a result, students do not encounter cases from a previous PCSE in which they were evaluated.

As noted, third- and fourth-year students take a single PSCE each semester with a portion of the students taking the first administration of the PCSE and others, the second administered that semester. Given this complication, we chose to focus on first- and second-year student performance for this study. During fall semester 2017 and spring semester 2018, four PCSEs were conducted as part of the SDC progress assessment. Second-year students from the class matriculating in 2016 and first-year students matriculating in 2017 completed the assessments. The scores in these four administrations of the PCSE for the two classes of students were used to assess growth in the students' clinical skills during the first 2 years of the curriculum and the psychometric characteristics of the PCSE.

## Generalizability study

We conducted a generalizability analysis of the PCSE domain scores separately for first- and second-year students. Students were the object of measurement and we considered SP cases as the only facet in the universe of admissible observations. Cases and students were crossed in the design.

Clearly other potential sources of measurement error exist beyond SP cases. For example, SPs can vary, both in their portrayal of the case and grading of the checklists; Another example of potential measurement error is variability in faculty grading of the post encounter station material. Given the data were collected from an actual examination that is used to test hundreds of students within a few weeks at two testing centers, building

in a balanced design that could adequately assess these and other potential sources of measurement error was not practical. As noted, the SP program expends a great deal of effort to ensure the cases are portrayed and the checklists graded consistently and accurately. The same is true for the faculty graded assessments of the post encounter stations were each faculty's rating are reviewed by the Director of Assessment who is a senior physician faculty member. The variability in performance of examinees managing different clinical cases has been noted in numerous studies going back to the late 1970's when it was labeled by Elstein and his colleagues as case specificity (*Elstein, Shulman & Sprakfa, 1978*). Given the high cost and logistical challenges of an incremental increase in the number of cases as a means of improving the reproducibility of PCSE scores, we felt assessing this source of measurement error was the most critical in factor to assess in this study.

As noted above, we are interested in both cross-sectional comparisons of the first- and second-year students' performance taking the same PCSE as well as the longitudinal growth of the students' performance across multiple administrations of the PCSE each containing a different set of SP cases. These two types of comparisons have different generalizability coefficients, and standard errors of measurement (*Cronbach et al., 1972*; *Brennan, 2001*). In the cross-sectional comparisons, the students at each level of training are assessed on the same eight SP cases. The error variance for comparing students is equal to the residual variance in the ANOVA design and equivalent to the error variance as defined in classical test theory (*Magnusson, 1967*). When making longitudinal comparisons of students over multiple examinations, the comparisons are based on different SP cases that are not perfectly parallel. As such, longitudinal comparisons include an additional source of error from the variation among different SP cases, have lower generalizability and larger standard errors of measurement than cross-sectional comparisons. The difference between these two types of measurement is often referred to as "norm-referenced" and "domain-referenced" measurement (*Brennan & Kane, 1977*).

We used GENOVA for conducting the generalizability analyses (*Crick & Brennan, 1983*). Details of how the variance components, standard errors and generalizability coefficients are estimated are contained in the appendences of *Brennan (2001)*. As noted, PCSE scores are reported as the percentage of possible points a student achieves in the domain across all eight cases. Since the generalizability analysis is based on case-level data, we conducted the generalizability analysis on the number of points achieved for each case. Since the generalizability coefficients are ratios of the expected values of variance components, the difference in metric did not impact the generalizability coefficients. It did, however, impact the standard error of measurement provided by GENOVA. To avoid this problem, we calculated standard errors of measurement from the observed standard deviation of the domain scores in the full examination and the generalizability coefficients using a common formula in classical test theory and describe in *Magnusson (1967)*.

We conducted the analysis separately for first- and second-year students. Since there was no easy way to combine the data from multiple administrations of the PCSE to conduct the generalizability analysis, we conducted the analysis on a single administration of the PCSE using the data from the first administration of the PCSE given in spring

semester 2018. While not ideal since the analysis was conducted with less data hence more susceptible to sampling error, it still provided estimates of the variance associated with students, cases and the residual necessary to estimate generalizability coefficients.

### Repeated measures analysis

To assess growth both cross-sectionally and longitudinally as well as their interaction, the data from both classes and the four administrations of the PCSE given over fall 2017 and spring 2018 semesters were analyzed using repeated measures ANOVA. The two classes of students formed the design over subjects and the four administrations of the PCSE formed the design over measures. Orthogonal polynomial contrasts were used to assess the shape of the growth curve over the four administrations of the PCSE. This type of analysis uses contrasts that partition the change in scores over the four administrations into a linear component that reflects the consistent (linear) change over administrations, a quadratic component that assesses the acceleration or deacceleration in the change over administrations and a cubic component that assesses the change in acceleration/deacceleration over the administrations of the PCSE. The repeated measures analysis and the generation of summary statistics was done using SPSS Version 25. We considered ($p < 0.01$) as statistically significant for contrasts.

### Human subject protection

The student performance data and matriculation class were provided to the researchers in a deidentified format by the Office of Medical Education Research and Development honest broker. Given the PCSE was administered as a normal part of the SDC student evaluation program and the student data were deidentified by the recognized honest broker within the medical school, the data used in the study are not considered to be human subjects data by the Michigan State University Human Research Protection Program (*Honest Broker for Educational Scholarship, 2005*).

## RESULTS

There was complete data over the four administrations of the PCSE for 183 first-year and 170 second-year students. Table 1 presents the means and standard deviations for each of the six domains in each class over the four administrations of the PCSE. Figure 1 presents the mean performance for each class across the four administrations in a graphical format. Table 2 provides the generalizability coefficients and estimated standard errors of measurement for cross-sectional (norm referenced) and longitudinal (domain referenced) comparisons. These coefficients are essentially a "D" study (*Brennan, 2001*) based on the examinations as administered with eight cases. As noted above, the generalizability analysis was based on the first administration of the PCSE spring term. A compressed archive including the GENOVA outputs for each domain/year combination that includes sums of squares, variance component estimates, and other details of the generalizability analysis is included in the Supplemental Material.

As can be seen in Table 2, physical examination and the post-encounter stations had the highest generalizability coefficients at around 0.50 for cross-sectional comparisons.

**Table 1 Percentage of ratings/checklist items correct by domain and exam.**

| Matriculation year | | Fall 1 2017 | Fall 2 2017 | Spring 1 2018 | Spring 2 2018 |
|---|---|---|---|---|---|
| Interactive skills | | | | | |
| First year | Mean | 85.48 | 84.93 | 91.01 | 92.32 |
| | SD | 10.62 | 7.51 | 6.08 | 4.81 |
| Second year | Mean | 91.52 | 91.08 | 91.96 | 92.88 |
| | SD | 7.70 | 5.80 | 4.67 | 4.65 |
| History | | | | | |
| First year | Mean | 31.28 | 55.50 | 68.14 | 64.52 |
| | SD | 11.76 | 10.75 | 8.13 | 8.13 |
| Second year | Mean | 63.76 | 74.98 | 73.85 | 70.44 |
| | SD | 10.47 | 5.78 | 8.22 | 7.03 |
| Physical examination | | | | | |
| First year | Mean | 24.01 | 38.16 | 40.26 | 51.32 |
| | SD | 7.94 | 11.71 | 11.07 | 10.80 |
| Second year | Mean | 52.01 | 61.06 | 58.79 | 60.81 |
| | SD | 10.37 | 9.15 | 9.74 | 9.49 |
| Counseling | | | | | |
| First year | Mean | 57.23 | 62.02 | 62.22 | 61.39 |
| | SD | 11.37 | 11.38 | 12.24 | 10.21 |
| Second year | Mean | 73.26 | 71.48 | 68.15 | 64.28 |
| | SD | 10.75 | 9.13 | 8.67 | 8.77 |
| Safety | | | | | |
| First year | Mean | 72.41 | 45.90 | 72.30 | 83.66 |
| | SD | 9.72 | 8.42 | 9.47 | 6.75 |
| Second year | Mean | 86.96 | 54.94 | 77.81 | 85.68 |
| | SD | 5.53 | 6.49 | 9.02 | 6.83 |
| Post encounter | | | | | |
| First year | Mean | 37.26 | 29.64 | 56.10 | 46.04 |
| | SD | 8.20 | 5.79 | 7.20 | 6.91 |
| Second year | Mean | 57.49 | 47.28 | 68.21 | 55.59 |
| | SD | 7.23 | 8.63 | 6.97 | 6.60 |

The generalizability coefficients were the lowest for second-year students in the patient interaction domain and in the safety domain for both first- and second-year students.

In the repeated measures analysis, the main effect for medical school year, was significant ($p < 0.001$) for all six domains. As can be seen in Table 1 and Fig. 1, the second-year students outperformed first-year students in all six domains. The main effects for administration (linear, quadratic, and cubic) were also significant ($p < 0.001$). It appears that case difficulty significantly impacted on the change in scores from administration to administration. There was a statistically significant ($p < 0.001$) interaction between year and linear change over administrations for all six domains. The quadratic and cubic components also were significant for the history domain

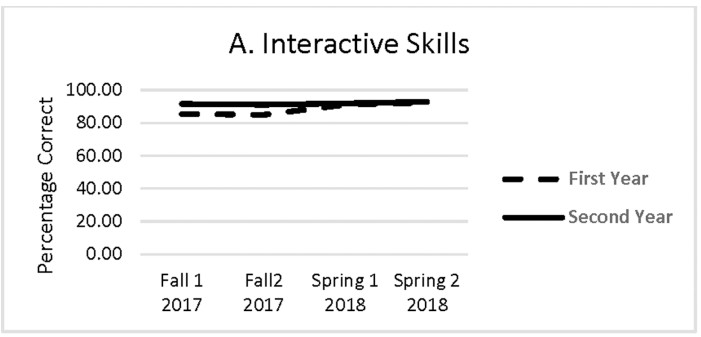

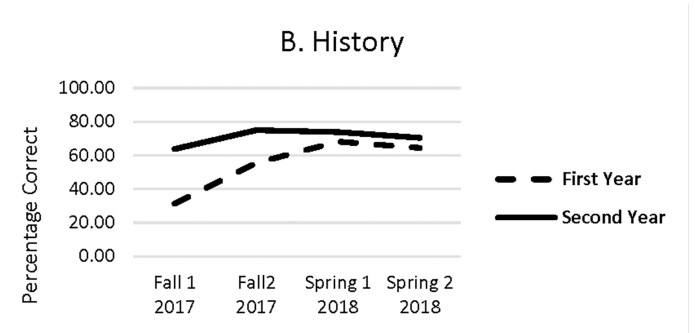

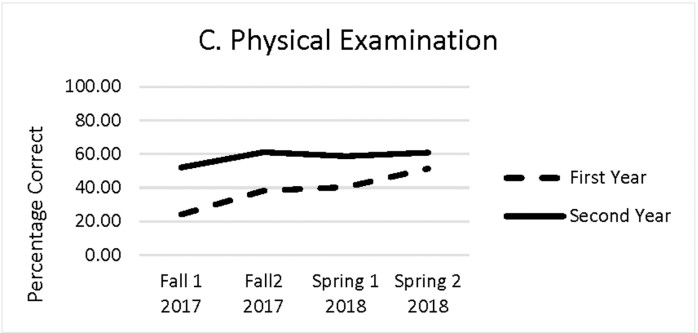

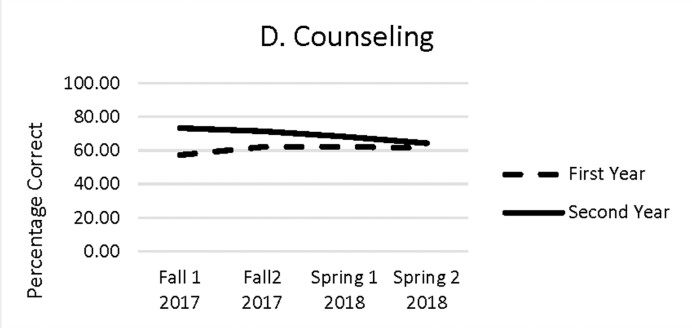

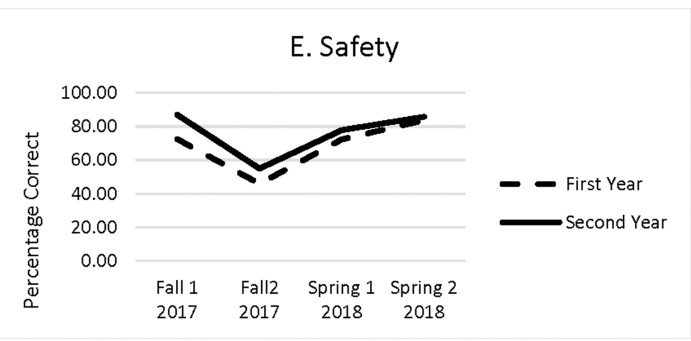

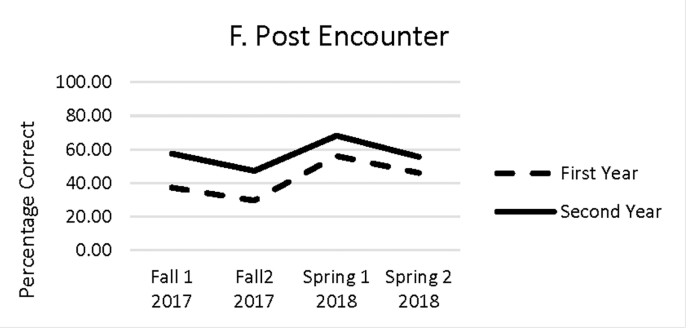

**Figure 1 Growth in six clinical skills domains among first- and second-year medical students.** (A) Interactive Skills, (B) History, (C) Physical Examination, (D) Counseling, (E) Safety, (F) Post Encounter. Percentage of possible points in the six clinical skill domains for first-and second-year students on four examination given over two semesters.

($p < 0.001$). As can be seen in Fig. 1, the gap in performance between the first- and second-year students narrowed across the four administrations for all six domains. The full results of the repeated measures analysis displayed by SPSS are contained in a compressed archive in the Supplemental Materials.

## DISCUSSION

Progress clinical skills examinations offer many of the benefits of written progress testing while assessing skills that cannot be measured via written examinations. Unfortunately, measurement error in the form of case specificity is a significant challenge when implementing a PCSE program, particularly for assessing longitudinal growth. We found the generalizability of the PCSE domain scores as our examination is currently configured to be considerably lower than is generally acceptable for high stakes examinations.

**Table 2 Generalizability coefficients and standard errors of measurement.**

| Skill domain | Level | Observed St. Dev. | Generalizability | | Standard error | |
|---|---|---|---|---|---|---|
| | | | Norm | Domain | Norm | Domain |
| | | | Referenced | | Referenced | |
| Patient interaction | MS-1 | 6.08 | 0.39 | 0.35 | 4.75 | 4.90 |
| | MS-2 | 4.67 | 0.08 | 0.07 | 4.48 | 4.50 |
| Hypothesis driven history | MS-1 | 8.13 | 0.27 | 0.17 | 6.95 | 7.41 |
| | MS-2 | 7.03 | 0.26 | 0.16 | 6.05 | 6.45 |
| Physical examination | MS-1 | 11.07 | 0.54 | 0.43 | 7.51 | 8.36 |
| | MS-2 | 9.74 | 0.45 | 0.36 | 7.22 | 7.79 |
| Counseling | MS-1 | 12.24 | 0.44 | 0.20 | 9.16 | 10.95 |
| | MS-2 | 8.67 | 0.27 | 0.08 | 7.41 | 8.32 |
| Safety | MS-1 | 9.47 | 0.08 | 0.04 | 9.08 | 9.28 |
| | MS-2 | 9.02 | 0.04 | 0.02 | 8.84 | 8.93 |
| Post encounter | MS-1 | 7.20 | 0.51 | 0.23 | 5.04 | 6.32 |
| | MS-2 | 6.97 | 0.43 | 0.19 | 5.26 | 6.28 |

**Note:**
The generalizability coefficients, observed standard deviations and the estimated standard errors are based on first administration of the PCSE Spring Term 2018.

The extremely low generalizability coefficients for the patient interaction domain for the second-year students appears to be due mainly to a ceiling effect. Students in the SDC have largely mastered these skills by the end of the first year of the curriculum. At that point, students on average achieve over 90% of the possible points in this domain. The variability in the scores that is left appears to be mostly error variance. This, in of itself, is not necessarily a problem. It means we cannot easily differentiate among second-year students in their ability to communicate with patients because, for the most part, they have mastered this skill domain and what little difference there is in the scores does not replicate from case to case.

There is general agreement that ensuring patient safety and avoiding potential medical errors is a very important focus in medical training. These skills often cannot be adequately assessed through written examinations. We believe our PCSE is the first clinical skills examination to break these skills out as a specific competency domain that is scored separately from other clinical skills domains. From a curricular standpoint focusing on safety as a clinical skill domain make sense. From a psychometric perspective, it appears that safety, at least as we have conceptualized it in the PCSE, is not a unidimensional construct. This does not mean that safety is any less important. It just brings into question whether it is possible to treat safety as we have defined it as a unidimensional clinical skill domain. Clearly, more research is needed on how to conceptualize safety. It appears we may have defined safety too broadly, and there may be multiple unidimensional clusters of skills that we are currently conceptualizing under the broad rubric of safety.

Measurement error in classical test theory is assumed to be random with an expectation of zero (*Magnusson, 1967*). While this source of residual measurement error remains a

significant problem for assessing the performance of individual students using PCSE, it is less of a problem for assessing groups of students for in research and evaluation since the error in individual student scores tends to cancel out when averaged over multiple students in a research or evaluation study. This is not the case for the error associated with measuring longitudinal growth over multiple PCSE administrations using different sets of cases. The difficulty of the cases in different administrations of the PCSE is confounded with student growth in performance, making it difficult to assess longitudinal growth in clinical skills. Unfortunately, assessing longitudinal growth one of the important benefits of progress testing.

The interaction between longitudinal growth and level of training is not directly subject to this type of measurement error. The repeated measures analysis demonstrated that there was a statistically and we feel educationally significant interaction between level of training and linear growth in all six domains. As can be seen in Table 1 and Fig. 1, the difference in the scores between first- and second-year students narrowed in successive administrations of the PCSE. In other words, first-year students' growth in all six clinical skills domains exceeded the second-year students' growth. It is not clear if this was due to participation in authentic clinical experiences early in their training, other extensive clinical skills training the students received, or the combination of both. Clearly a focus on early clinic training results in rapid growth in clinical skills over the first year of the curriculum. We expect the second-year students' clinical skills are also improving, but due to variability in difficulty among the different cases used in each administration of the examination, it is difficult to measure.

Although the generalizability of the PCSE domains is low, the results are consistent with earlier research on clinical skills examinations. Prior studies found similar generalizability coefficients for data gathering skills (*Swanson & Norcini, 1989*; *Clauser et al., 2009*) and communication skills (*Hodges et al., 1996*). As noted by *Swanson & Norcini (1989)*, clinical skills examinations given in medical schools, like our PCSE, are typically about 2 hours in length. Extrapolating from the generalizability coefficients observed in these studies and ours, the examination time needed to obtain generalizability coefficients considered appropriate for high stakes decisions would be about 8 h. Given logistical and resources constraints, conducting 8 hour clinical skills examinations in medical schools is not feasible in most situations including our PCSE.

*Swanson & Norcini (1989)* suggest focusing on pass/fail decisions and using sequential testing as a means of making more accurate decisions about the competency of students when the reproducibility of clinical skills examinations is limited as in the PCSE. Our approach to this problem is similar to what Swanson and Norcini suggest. We have used a rigorous standard setting process to set the minimum pass requirements for students for each semester of the curriculum. First- and second-year students can receive a passing grade if they meet these requirements on either or both PCSEs given in a semester. Third- and fourth-year students, who normally only take the examination once a semester, can take a makeup examination if they perform below the requirement and are able to pass the semester if they achieve the passing requirements on the makeup examination.

## LIMITATIONS

As noted above, this study has several limitations. Due to design constraints, we were unable to assess several potential sources of measurement error in the generalizability study. Administering a PCSE to all the students in a medical school in a short period of time at regular intervals over the curriculum is extremely resource intensive. The costs and resources necessary to conduct a PCSE program is largely proportional to the number of cases in the examination. For this reason, and the fact numerous studies have shown case specificity is a major source of measurement error in clinical skills examinations, the information provided by the generalizability study is very valuable for our program and potentially other medical schools despite the limitations. Like in any small-scale study, however, readers need to be cautious in extrapolating the results of the generalizability study to other examinations that differ significantly from our PCSE.

As noted, while it is clear the students' clinical skills grew rapidly during the first year of the curriculum narrowing the gap with their second year counterparts, unfortunately due to case specificity it is difficult to ascertain the true growth curve in either class. Readers also need to be cautious in extrapolating our findings to other medical schools that differ significantly from ours in the structure of their preclinical curricula even when they include authentic clinical experiences.

## CONCLUSIONS

Progress clinical skills examinations provide a standardized methodology for assessing important clinical skills that cannot be evaluated by written examinations. As in previous research on clinical skills examinations, we found that case specificity and random measurement error are major impediments for using PCSE scores to make high stakes decisions about student competency. While the situation is not ideal, this limitation can be addressed by focusing on pass/fail decisions based on rigorous standard setting and using sequential testing to help ensure the accuracy of decisions about student competency based on these examinations.

One of the advantages of progress testing is the ability to assess longitudinal growth in knowledge and skills over a course of training. Unfortunately, the variability in case difficulty among different administrations of the PCSE limited our ability to directly assess longitudinal growth. Over time, as we develop a pool of cases that have been used previously in the PCSE, hopefully we will be able to use the data from previous administrations to balance out case difficulty in each administration of the PCSE and be in a better position to assess longitudinal growth in clinical skills.

While measurement error associated with using different cases makes it difficult to directly assess longitudinal growth in clinical skills, we did find that first-year students in the SDC gain basic clinical skills rapidly, narrowing the gap in skill level with their second-year counterparts. This study suggests implementing regular clinical skills training and early authentic clinical experiences results in the rapid growth of clinical skills over the first year of medical school.

Ensuring patient safety and reducing the chances of serious medical errors are important curricular goals in both undergraduate and graduate medical training. Clinical skills examinations offer a standardized means for assessing actions that help ensure patient safety. We defined safety broadly in our examination, and it appears that the safety domain as currently designed is not a unidimensional construct. Further research is needed on how to conceptualize safety and create reproducible measures of this important aspect of medical training.

## ACKNOWLEDGEMENTS

Ann Taft maintains the student evaluation database used in the study and acted as the honest broker, creating a deidentified research dataset for use in the study. The authors greatly appreciate her help.

### Funding

The authors received no funding for this work.

### Competing Interests

The authors declare that they have no competing interests.

### Author Contributions

- Heather S. Laird-Fick conceived and designed the experiments, performed the experiments, authored or reviewed drafts of the paper, and approved the final draft.
- Chi Chang analyzed the data, authored or reviewed drafts of the paper, and approved the final draft.
- Ling Wang analyzed the data, authored or reviewed drafts of the paper, and approved the final draft.
- Carol Parker analyzed the data, authored or reviewed drafts of the paper, and approved the final draft.
- Robert Malinowski conceived and designed the experiments, performed the experiments, authored or reviewed drafts of the paper, and approved the final draft.
- Matthew Emery conceived and designed the experiments, performed the experiments, authored or reviewed drafts of the paper, and approved the final draft.
- David J. Solomon analyzed the data, prepared figures and/or tables, authored or reviewed drafts of the paper, and approved the final draft.

### Human Ethics

The following information was supplied relating to ethical approvals (i.e., approving body and any reference numbers):

The student performance data and matriculation class were provided to the researchers in a deidentified format by the Office of Medical Education Research and Development (OMERAD) honest broker. Given the PCSE was administered as a normal part of the SDC student evaluation program and the student data were deidentified by the recognized
honest broker within the medical school, the data used in the study are not considered to be human subjects data by the Michigan State University Human Research Protection Program.

https://omerad.msu.edu/research/honest-broker-for-educational-scholarship

## Data Availability

The raw data and data description are available in the Supplemental File.

## Supplemental Information

Supplemental information for this article can be found online at http://dx.doi.org/10.7717/peerj.9091#supplemental-information.

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
