# Peer review of "Assessing the growth in clinical skills using a progress clinical skills examination"

_PeerJ, doi:10.7717/peerj.9091_

## Round 0.1 · original submission · Major Revisions

Your manuscript has been reviewed and requires modifications prior to making a decision. All reviewers have found multiple serious flaws in your study. Findings of the study were not clearly reported, the study design and statistical methods need much improvement. The comments of the reviewers are included at the bottom of this letter. I agree with the evaluation and I would, therefore, request for the manuscript to be revised accordingly.

Reviewer 1 ·

Basic reporting

Basic reporting is excellent with professional English used throughout. Literature references are appropriate and article is generally well structured. Figures and tables are concise and relevant.

Experimental design

Generally well described in terms of implementation of the PCSE program. However, a few elements should be addressed:

1) The research question is not strongly defined. It is stated that this is a generalizability study of PSCE testing. However it is not emphasized why generalizability is important, and does not define generalizability coefficients (and what are acceptable ranges of these coefficients) for an audience that may be unfamiliar with this concept.

2) It is stated in the result section that "The main effects for administration (linear,
quadratic, and cubic) were also significant (p < 0.001)". These terms should be defined in the methods section especially for for an audience that may be unfamiliar with these concepts.

3) It is stated that students are assessed over six domains. However these domains are not clearly defined in the methods section. It is stated "Each encounter assesses some combination of <patient interaction skills>, <hypothesis-driven history gathering>, <physical examination>, <counseling> and <safety behaviors> using checklist and rating items completed by the SP. Post-encounter tasks, which are graded by faculty members, assess the <application of medical knowledge>, <clinical reasoning>, and <clinical documentation>." To me, this appears like eight domains (I have enclosed each perceived domain in <brackets>). It would be better if each of the six domains was separated in a numbered list.

Validity of the findings

Generally well reported findings with some areas that need clarification:

1) It is stated that "Table 2 provides a summary of the generalizability coefficients and estimated standard errors of measurement for cross-sectional and longitudinal comparisons". However Table 2 only provides information about the cross-sectional comparison from Spring 2018. It does not provide any longitudinal comparisons.

In fact, this is reflected in the prior methods section when it is stated "we conducted the generalizability analysis on a single administration of the PCSE and used the data from the first administration of the PCSE given in spring semester 2018 for conducting the generalizability analysis."


2) In the discussion, it is stated "Physical examination and the post-encounter stations had the highest generalizability coefficients at around 0.50 for cross-sectional comparisons. The generalizability coefficients were the lowest for second-year students in the patient interaction domain and in the safety domain for both first- and second-year students." These sentences should be part of the results section, not the discussion section.

3) In the discussion, it is stated that safety is assessed by "proper identification of patient and visitors; infection prevention/control; medication safety; handovers; conflict resolution; team, communication; informed consent; risk identification; open disclosure of adverse events; patient and family engagement". This detailed description should be in the methods section, not in the discussion. In fact, each of the six domains should be described with more detail like this in the methods section (perhaps when separated in a numbered list, like I have previously mentioned).

Reviewer 2 ·

Basic reporting

Although the authors provided an Introduction and background of the study, literature review is scanty or inadequate, with only 16 references. Only 10 out of the 16 references are journal articles, with only 5 of them published in 2010 or later. The author’s claim that there has been relatively little research done on PCSEs should be supported by more recent literature from journal articles.
Although raw data from the study were provided as a supplementary file, findings of the study were not clearly reported. In the Results section, two tables and one figure were presented. However, Table 1 and Figure 1 appear to provide the same data, except that Figure 1 is in the graphical format. The information displayed in Table 2 needs to be more clearly described or explained in the text. For the repeated measures ANOVA, the authors just described the results in the text without referring to any output of the analysis or any table(s) summarising the findings, in particular the main effects and interaction. Furthermore, in the second paragraph of the Results section, descriptive statistics (such as in Table 1 and Figure 1) and inferential statistics (such as repeated measures analysis) were combined. This is confusing to readers.

Experimental design

Although the research falls within the scope of the journal, no research questions or hypotheses were identified in this report. Only two study objectives were found in lines 80 to 85. No gap in research, either theory or practice, was clearly identified and it is not clear how the research fills the knowledge gap.
For both the generalizability (G) study and repeated measures ANOVA, the methods should be described with sufficient detail and clarity to enable replication studies to be conducted. The authors devoted three pages of the manuscript to the section on Materials and Methods. However, the G study was not clearly described, and terminology in G study was not used. The description from lines 149 to 152 was not clear. For example, “We considered standardized patient case as the only facet in the universe of admissible observations (UAO)”. Was it referring to SP or case? Please also define the set of all facets that constitute the UAO.
With regards to the G study, the authors should have identified and/or defined the following:
1. What are the potential sources of measurement error (facets)?
2. What is the study design? Is it a one-facet design/two-facet design/multiple facet design?
3. What is the object of measurement (facet of differentiation)?
4. What are the sources of variance (facets of generalization)?
5. Is it a crossed design or nested design?
6. Is it a one-factor repeated measure ANOVA or a two-factor repeated measures ANOVA…?
The researchers/authors should justify any decision about the inclusion of facet(s), and provide supporting evidence about the inclusion of each facet to the consistency and accuracy of the measurement procedure. Statistical method like generalizability (G) theory is only useful if it is accompanied by careful experimental design, planning and analysis (Bloch & Norman, 2012).
The following references could be helpful:
• Bloch R, Norman G. (2012). Generalizability theory for the perplexed: a practical introduction and guide: AMEE Guide No. 68. Med Teach. 2012; 34(11): 960-92. doi: 10.3109/0142159X.2012.703791.
• Mohsen Tavakol, Robert L. Brennan. (2013). Medical education assessment: a brief overview of concepts in generalizability theory. International Journal of Medical Education; 4: 221-222. doi: 10.5116/ijme.5278.a850221
It was only mentioned that GENOVA was the software package used to perform generalizability analysis. However, computation of G coefficients was also not explained.

Validity of the findings

As repeated measures ANOVA is an experimental design, both internal validity and external validity of the study should be considered and discussed. However, validity of the findings was not reported.

Additional comments

A good attempt at using the generalizability theory instead of the classical test theory to examine reliability of clinical assessment such as OSCE. However, more recent and adequate literature is needed. The study design needs much improvement too. Clarity of the writing needs to be enhanced.

Reviewer 3 ·

Basic reporting

This is an interesting study and addresses a topic of great importance for Medical Education. However, it fails to meet the objective as there is no growth in many competencies (sometimes even decreasing).

Experimental design

I have some doubts about the results of the generalilizability.

Since the goal in a G study is to estimate as many sources of variance as are potentially relevant in order to identify the major sources of measurement error, I ask:

Couldn't the curricular year being included in the analysis as main effect instead of being conducted separately?

Isn't it important to present the variance component for the student and residual variance?

There is no description of the Table 2, which is one of the main results of the study. Does the variance correspond to % of the total variance? If not, how do we know if it is low or high?

Validity of the findings

How can generalilizability be improved? Why not propose a D study?

---

## Round 0.2 · accepted · Accept

The authors addressed the reviewer's concerns and substantially improved the content of MS.

So, based on my own assessment as an editor, no further revisions are required and the MS can be accepted in its current form.

Reviewer 1 ·

Basic reporting

Acceptable after the revisions I suggested previously thank you.

Experimental design

Acceptable after the revisions I suggested previously thank you.

Validity of the findings

Acceptable after the revisions I suggested previously thank you.